# Risk assessment models for venous thromboembolism in pregnancy and in the puerperium: a systematic review

Abdullah Pandor [1], Jahnavi Daru [2], Beverley J Hunt [3], Gill Rooney [1], Jean Hamilton [1], Mark Clowes [1], Steve Goodacre [1], Catherine Nelson-Piercy [3], Sarah Davis [1]

## ABSTRACT

**Objectives** To assess the comparative accuracy of risk assessment models (RAMs) to identify women during pregnancy and the early postnatal period who are at increased risk of venous thromboembolism (VTE).

**Design** Systematic review following Preferred Reporting Items for Systematic Reviews and Meta-Analyses guidelines.

**Data sources** MEDLINE, Embase, Cochrane Library and two research registers were searched until February 2021.

**Eligibility criteria** All validation studies that examined the accuracy of a multivariable RAM (or scoring system) for predicting the risk of developing VTE in women who are pregnant or in the puerperium (within 6 weeks post-delivery).

**Data extraction and synthesis** Two authors independently selected and extracted data. Risk of bias was appraised using PROBAST (Prediction model Risk Of Bias ASsessment Tool). Data were synthesised without meta-analysis.

**Results** Seventeen studies, comprising 19 externally validated RAMs and 1 internally validated model, met the inclusion criteria. The most widely evaluated RAMs were the Royal College of Obstetricians and Gynaecologists guidelines (six studies), American College of Obstetricians and Gynecologists guidelines (two studies), Swedish Society of Obstetrics and Gynecology guidelines (two studies) and the Lyon score (two studies). In general, estimates of sensitivity and specificity were highly variable with sensitivity estimates ranging from 0% to 100% for RAMs that were applied to antepartum women to predict antepartum or postpartum VTE and 0% to 100% for RAMs applied postpartum to predict postpartum VTE. Specificity estimates were similarly diverse ranging from 28% to 98% and 5% to 100%, respectively.

**Conclusions** Available data suggest that external validation studies have weak designs and limited generalisability, so estimates of prognostic accuracy are very uncertain.

**PROSPERO registration number** CRD42020221094.

¹ScHARR, The University of Sheffield, Sheffield, UK
²Institute of Population Health Sciences, Queen Mary University of London, London, UK
³Guy's and St Thomas' NHS Foundation Trust, London, UK

**Correspondence to**
Abdullah Pandor;
a.pandor@sheffield.ac.uk

## STRENGTHS AND LIMITATIONS OF THIS STUDY

⇒ A number of risk assessment models for venous thromboembolism (VTE) in pregnancy and puerperium have been developed using a variety of methods and based on a variety of predictor variables.

⇒ This systematic review provides a comprehensive review of risk assessment models for predicting the risk of developing VTE in women who are pregnant or in the puerperium (within 6 weeks post-delivery).

⇒ The newly developed PROBAST (Prediction model Risk Of Bias ASsessment Tool) was used to evaluate the risk of bias and applicability of the available evidence.

⇒ Heterogeneity in the included studies (participants, inclusion criteria, clinical condition, outcome definition and measurement) and variable reporting of items precluded meta-analysis.

⇒ Limitations of the existing evidence and areas of future research are highlighted.

## INTRODUCTION

Venous thromboembolism (VTE) remains an important cause of maternal morbidity and mortality in the developed world.[1] While uncommon, VTE complications can occur at a rate of 1–2 per 1000 deliveries and can develop at any time during pregnancy.[2–4] The risks substantially increase during the postpartum period (6 weeks post-delivery)[5] and can be as high as 60-fold in some individuals compared with age-matched non-pregnant women.[6] Preventative treatment with low-dose anticoagulation (thromboprophylaxis) has the potential to reduce the risk of symptomatic and asymptomatic VTE in pregnancy and the postpartum period.[5] Consequently, various prominent international guidelines recommend targeted thromboprophylaxis for pregnant and puerperal women deemed to be at high risk of VTE.[5 7–13] However, these expert-based consensus guidelines vary substantially with regards to the threshold of risk (based on certain risk factors) and the timing, dose and duration of pharmacological thromboprophylaxis.

Risk assessment models (RAMs) have been developed to help stratify the risk of VTE during pregnancy and the early

postnatal period. These models use clinical information from the patient's history and examination to identify those with an increased risk of developing VTE who are most likely to benefit from pharmacological thromboprophylaxis. Inappropriate use of VTE prophylaxis may not reduce VTE rates and may cause unnecessary harm especially through bleeding and bruising.[14] While RAMs could improve the ratio of benefit to risk and benefit to cost, it is unclear which VTE RAM are best applied to guide decision-making for thromboprophylaxis in clinical practice and thereby optimise patient care.

The aim of this systematic review was to identify primary validation studies and determine the accuracy of individual RAMs that identify pregnant and postpartum women at increased risk of developing VTE who could be selected for thromboprophylaxis.

## METHODS

A systematic review was undertaken in accordance with the general principles recommended in the Preferred Reporting Items for Systematic Reviews and Meta-Analyses (PRISMA) statement.[15] This review was part of a larger project on Thromboprophylaxis in pregnancy and after delivery[16] and was registered on the International Prospective Register of Systematic Reviews (PROSPERO) database.

### Eligibility criteria

All studies evaluating the accuracy (eg, sensitivity, specificity, C-statistic) of a multivariable RAM (or scoring system) for predicting the risk of developing VTE were eligible for inclusion. We primarily sought and selected studies that included validation of the model in a group of patients that were not involved in the development of the prediction model. Although the included studies could have reported derivation of the model (for internal validation), we only used the external validation data to estimate accuracy, where appropriate. The study population of interest in our review consisted of pregnant and postpartum (within 6 weeks post-delivery) women who are at increased risk of developing a VTE and receiving care in both hospital, community and primary care settings. Studies that focused on non-pregnant women were excluded as these patient groups have VTE risk profiles that differ markedly from the obstetric population.

### Data sources and searches

Potentially relevant studies were identified through searches of several electronic databases and research registers. This included MEDLINE (OvidSP from 1946), Embase (OvidSP from 1974), the Cochrane Library (https://www.cochranelibrary.com from inception), ClinicalTrials.gov (US National Institutes of Health from 2000) and the International Clinical Trials Registry Platform (WHO from 1990). All searches were conducted from inception to February 2021. The search strategy used free text and thesaurus terms and combined synonyms relating to the condition (eg, VTE in pregnant and postpartum women) with risk prediction modelling terms.[17] No language or date restrictions were used. Searches were supplemented by hand-searching the reference lists of all relevant studies (including existing systematic reviews); forward citation searching of included studies; contacting key experts in the field; and undertaking targeted searches of the World Wide Web using the Google search engine. Further details on the search strategy can be found in the online supplemental appendix S1.

### Study selection

All titles were examined for inclusion by one reviewer (GR) and any citations that clearly did not meet the inclusion criteria (eg, non-human, unrelated to VTE in pregnancy and the puerperium) were excluded (for quality assurance a random subset of 20% was checked by a second reviewer (AP)). All abstracts and full-text articles were then examined independently by two reviewers (GR and AP). Any disagreements in the selection process were resolved through discussion or if necessary, arbitration by a third reviewer (JD) or the wider group (BJH, CN-P, SG) and included by consensus.

### Data extraction and quality assessment

For eligible studies, data relating to study design, methodological quality and outcomes were extracted by one reviewer (GR) into a standardised data extraction form and independently checked for accuracy by a second reviewer (AP). Any discrepancies were resolved through discussion, or if this was unsuccessful, a third reviewer's opinion was sought (JD). Where multiple publications of the same study were identified, data were extracted and reported as a single study.

The methodological quality of each included study was assessed using PROBAST (Prediction model Risk Of Bias ASsessment Tool).[18 19] This instrument includes four key domains: participants (eg, study design and patient selection), predictors (eg, differences in definition and measurement of the predictors), outcome (eg, differences related to the definition and outcome assessment) and statistical analysis (eg, sample size, choice of analysis method and handling of missing data). Each domain is assessed in terms of risk of bias and the concern regarding applicability to the review (first three domains only). To guide the overall domain-level judgement about whether a study is at high, low or an unclear (in the event of insufficient data in the publication to answer the corresponding question) risk of bias, subdomains within each domain include several signalling questions to help judge with bias and applicability concerns. An overall risk of bias for each individual study was defined as low risk when all domains were judged as low; and high risk of bias when one or more domains were considered as high. Studies were assigned an unclear risk of bias if one or more domains were unclear, and all other domains were low.

## Data synthesis and analysis

Due to significant levels of heterogeneity between studies (study design, participants, inclusion criteria) and variable reporting of items, a meta-analysis was not considered possible. As a result, a prespecified narrative synthesis approach[20 21] was undertaken, with data being summarised in tables with accompanying narrative summaries that included a description of the included variables, statistical methods and performance measures (eg, sensitivity, specificity and C-statistic (a value between 0.7 to 0.8 and >0.8 indicated good and excellent discrimination, respectively; and values <0.7 were considered weak)),[22] where applicable. All analyses were conducted using Microsoft Excel 2010 (Microsoft Corporation, Redmond, Washington, USA).

## Patient and public involvement

Patients and the public were not involved in the design or conduct of this systematic review.

## RESULTS

### Study flow

Figure 1 summarises the process of identifying and selecting relevant literature. Of the 2268 citations identified, 16 studies[23–38] investigating 19 unique externally validated RAMs met the inclusion criteria. Only one of these studies[35] presented data on model development and external validation (this study used UK Clinical Practice Research Data linked to Hospital Episode Statistics to develop a risk prediction model and externally validated using Swedish medical birth registry data). The remaining studies focused on external validation with no description of the initial derivation methodology.[23–34 36–38] Due to the lack of model derivation studies with external validation, we also identified and included one internal validation study for completeness (ie, prediction model development without external validation).[39] This study used a bootstrap validation approach to capture optimism in model performance[40 41] when applied to similar future patients. Most of the full-text articles (n=97)

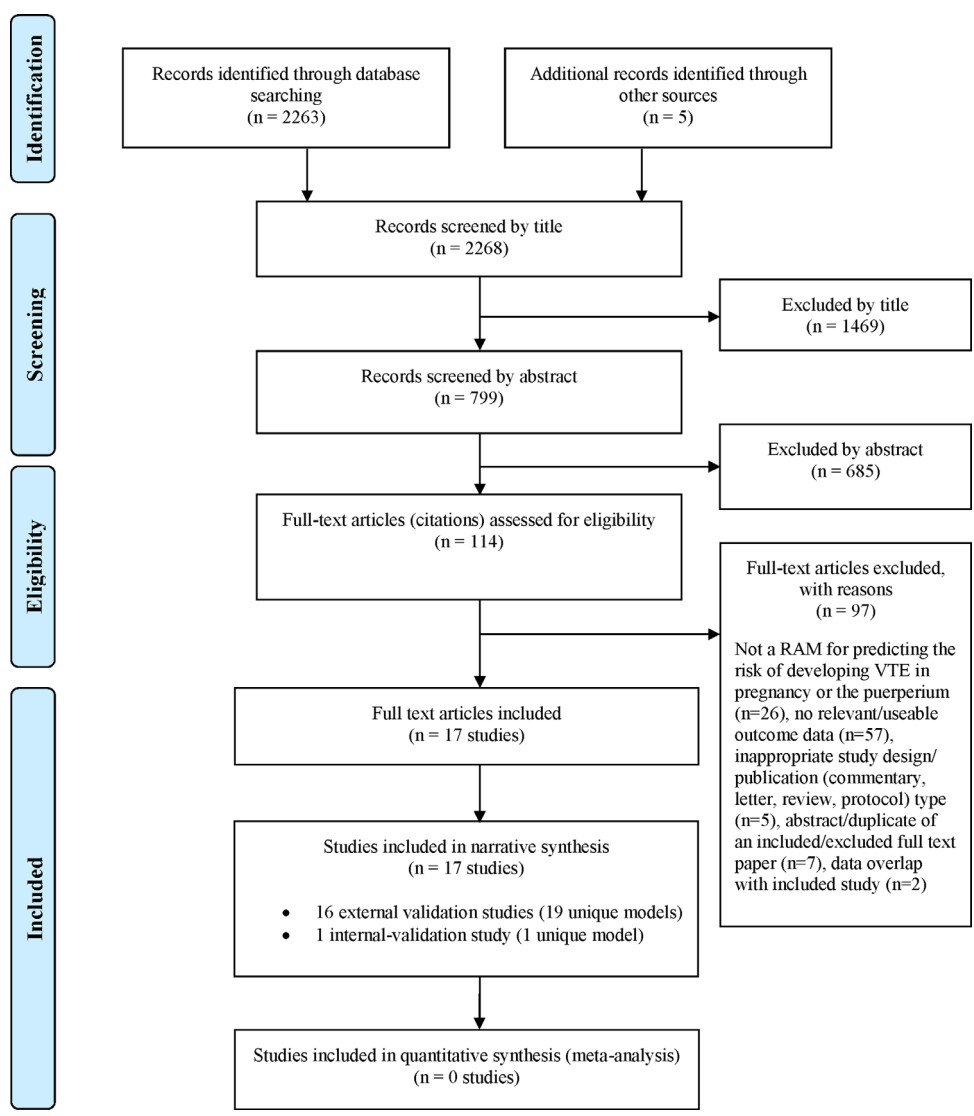

**Figure 1** Study flow chart (adapted). RAM, risk assessment model; VTE, venous thromboembolism.

were excluded primarily based on not using an RAM for predicting the risk of developing VTE during pregnancy or the puerperium, having no useable or relevant outcome data or an inappropriate study design (eg, reviews, commentaries or study protocols). A full list of excluded studies with reasons for exclusion is provided in online supplemental appendix S2.

### Study and patient characteristics

The design and participant characteristics of the 17 included studies are summarised in table 1. All studies were published between 2000 and 2020 and were undertaken in North America (n=4),[24 37–39] Southeast Asia (n=1),[33] Europe (n=10),[23 25–30 32 34 36] South America (n=1)[31] and one study was multicountry.[35] Sample sizes ranged from 52[31] to 662 387[35] patients in 14 observational cohort studies (6 prospective[25 27 28 31 33 36] (all single centre) and 8 retrospective[24 26 29 30 34 35 37 39] [2 of which were multicentre] in design). Sample sizes in two, single centre case–control studies[32 38] ranged from 76[38] to 2421[32] patients and one study used a non-randomised multicentre study design.[23] The mean age ranged from 27.8 years[39] to 34 years[25 29] (not reported in 7 studies).[24 27 32 34 36–38]

The majority of studies were conducted across antenatal and postnatal periods,[23 27–29 31 34 36 38] or postpartum period only[24–26 30 32 33 35 37 39] and generally included women at increased risk of VTE.[23–25 28 29 31–33 38 39] One study excluded women with a history of VTE[35] and six studies[26 27 30 34 36 37] included all pregnant women who delivered. Thromboprophylaxis was employed in about half (n=9)[23 25 28–31 33 35 36] of the studies, with the proportion receiving thromboprophylaxis ranging from 3%[35] to 100%.[23 28] The remaining studies did not report data on thromboprophylaxis use.

### VTE definition and case ascertainment

Only a few studies[23 27 32 36] defined the VTE endpoint (deep vein thrombosis and/or pulmonary embolism) as being confirmed by objective testing. Of the remainder, 3 studies[35 37 39] had no objective confirmation of VTE and 10 studies[24–26 28–31 33 34 38] did not report the methods for diagnosis confirmation. Although 9 studies[23 24 27 29 32–34 36 39] did not report the VTE risk period, the majority of the remaining studies used the RAMs to predict the occurrence of VTE up to 3 months after delivery.[25 28 30 31] Despite differences in study design, study participants, definitions, different criteria for the use of thromboprophylaxis and differences between doses of low molecular weight heparin (LMWH), the reported overall incidence of VTE in pregnancy and the puerperium was <1.3%.

### RAMs

The studies included in this review evaluated 19 externally validated RAMs[23–38] and 1 internally validated risk model.[39] While most RAMs focused solely on the estimate of thromboembolic risk, RAMs varied in design,

structure, threshold, dosage and duration for pharmacological prophylaxis. In addition, the individual predictors and their weighting varied markedly between RAMs. The most commonly used tools were the Royal College of Obstetricians and Gynaecologists guidelines (six studies),[24 30 33–35 37] American College of Obstetricians and Gynecologists (ACOG) guidelines (two studies),[30 33] Swedish Society of Obstetrics and Gynecology guidelines (two studies)[32 35] and the Lyon score (two studies).[28 29] A simplified summary of their associated characteristics and composite clinical variables is provided in online supplemental appendix S3.

### Risk of bias and applicability assessment

The overall methodological quality of the 17 included studies is summarised in table 2 and figure 2. The methodological quality of the included studies was variable, with most studies having high or unclear risk of bias in at least one item of the PROBAST. The main risk of bias limitations was related to patient selection factors (arising from retrospective data collection,[24 26 29 30 32 34 37–39] unclear exclusions/incomplete patient enrolment[24 26 27 31–34 36 38 39] or unclear criteria for patients receiving VTE prophylaxis)[23 30 35]; predictor and outcome bias (due to a general lack of details on the definition[24–26 28–31 33 34 38] and methods of outcome determination[24 26 28–31 33 34 37–39] and whether all predictors were available at the models intended time of use[23 24 29 31 32 34 36–39] or influenced by the outcome measurement)[23–28 30–39] and analysis factors (low event rates,[23–31 33–37 39] unclear handling of missing data[23–29 31–34 36–39] and failure in reporting relevant performance measures such as calibration and discrimination).[23–34 36–38]

Assessment of applicability to the review question led to the majority of studies being classed either as unclear (n=13)[23 26–30 32 34–39] or high (n=4)[24 25 31 33] risk of inapplicability. These assessments were generally related to patient selection (highly selected study populations, for example, selected women at increased risk of VTE, caesarean delivery only, single disease pathologies, single site settings), predictors (inconsistency in definition, assessment or timing of predictors) and outcome determination.

### Predictive performance of VTE RAMs (summary of results)

Table 3 and table 4 shows the sensitivity and specificity of RAMs that were applied to antepartum women to predict antepartum or postpartum VTE or applied postpartum to predict postpartum VTE, respectively, with the results grouped by RAM. However, any meaningful comparisons between these alone is difficult, without considering the models' corresponding discrimination and calibration metrics, which were not universally reported. Only one external validation study considered model discrimination and calibration. In this study by Sultan et al,[35] their recalibrated novel risk prediction model (also known as the Maternity Clot Risk) provided good discrimination and was able to discriminate postpartum women with

**Table 1** Study and population characteristics

| Author, year | Country | Design | Single/multicentre | Sample size | Population | Period | Mean age (years) | VTE prophylaxis | RAMs evaluated | Target condition, definition (risk period) | Incidence |
|---|---|---|---|---|---|---|---|---|---|---|---|
| Antepartum and postpartum following vaginal and caesarean delivery | | | | | | | | | | | |
| Bauersachs et al, 2007[23] | Germany | P, NRS | Multi | 810 | Women at increased risk of VTE (due to thromboembolic status and prior VTE) | March 1999 to December 2002 | 30.8 | 100% | ▲ EThIG | Antepartum and postpartum VTE, symptomatic (NR) | 0.62% (antepartum: 0.25%; postpartum: 0.37%) |
| Chauleur et al, 2008[27] | France | P, CS | Single | 2685 | All women who delivered | July 2002 to June 2003 | NR (median, 29) | NR | ▲ STRATHEGE | Antepartum and postpartum VTE (NR) | 0.34% (antepartum: 0.19%; postpartum: 0.15%) |
| Dargaud et al, 2017[28] | France | P, CS | Single | 445 | Women at increased risk of VTE (due to thromboembolic status and prior VTE) | January 2005 to January 2015 | 33 | 100% | ▲ Lyon | Antepartum and postpartum VTE, not defined (pregnancy and 3 months postpartum) | 1.35% |
| Dargaud et al, 2005[29] | France | R, CS | Single | 116 | Women at increased risk of VTE (due to thromboembolic status and prior VTE) | 2001 to 2003 | 34 | 53% | ▲ Lyon | Antepartum and postpartum VTE, not defined (NR) | 0.86% (antepartum only) |
| Hase et al, 2018[31] | Brazil | P, CS | Single | 52 | Hospitalised pregnant women with cancer | 1 December 2014 to 31 July 2016 | 31 | 57.7% | ▲ RCOG (modified) | Antepartum and postpartum VTE, not defined (pregnancy and 3 months postpartum) | Unable to estimate—no VTE |
| Shacaluga and Rayment, 2019 (correspondence)[34] | Wales | R, CS | Single | 42 000 | All managed pregnancies | 2009 to 2015 | NR | NR | ▲ All Wales ▲ RCOG | Antepartum and postpartum VTE, not defined (NR) | 0.08% (antepartum: 0.04%; postpartum: 0.04%) |
| Testa et al, 2015[36] | Italy | P, CS | Single | 1719 | All pregnant women enrolled in Pregnancy Healthcare Program | January 2008 to December 2010 | NR (median 33) | 4.6% | ▲ Novel (Testa) | Antepartum and postpartum VTE (NR) | Unable to estimate—no VTE |
| Weiss and Bernstein, 2000[38] | USA | CC | Single | 19 cases: 57 control* | Women with (confirmed cases) and without (unmatched control) VTE | 1987 to 1998 | NR | NR | ▲ Novel (Weiss) | Antepartum and postpartum VTE, not defined (pregnancy and 6 weeks postpartum) | – |
| Postpartum only following vaginal and caesarean delivery | | | | | | | | | | | |
| Chau et al, 2019[26] | France | R, CS | Single | 1069 (time period 2012: 557; 2015: 512) | All women who delivered | February to April 2012 and February to April 2015 | 2012: 29 2015: 29 | NR | ▲ Novel (Chau) | Postpartum VTE, not defined (8 weeks) | 2012: 0.18% 2015: 0.20% |
| Ellis-Kahana et al, 2020[39] † | USA | R, CS | Multi | 83 500 | All obese women (BMI >30 kg/m²) who delivered | 2002 to 2008 | 27.8 | NR | ▲ Novel (Ellis-Kahana) | Postpartum VTE (NR) | 0.13% |
| Gassmann et al, 2021[30] | Switzerland | R, CS‡ | Single | 344 | All women who delivered | 1–31 January 2019 | 32.2 | 24% | ▲ RCOG ▲ ACOG ▲ ACCP ▲ ASH | Postpartum VTE, not defined (3 months) | Unable to estimate—no VTE |

Continued

**Table 1** Continued

| Author, year | Country | Design | Single/ multicentre | Sample size | Population | Period | Mean age (years) | VTE prophylaxis | RAMs evaluated | Target condition, definition (risk period) | Incidence |
|---|---|---|---|---|---|---|---|---|---|---|---|
| Lindqvist et al, 2008[32] | Sweden | CC | Single | 37 cases: 2384 control | All women with (confirmed cases) and without (unselected population-based control) VTE | 1990 to 2005 | NR | NR | ▲ SFOG (Swedish guidelines) | Postpartum VTE (NR) | – |
| Sultan et al, 2016[35] | England (derivation)§ and, Sweden (validation) | R, CS | Multi | 662387 (validation cohort)§ | All women (with no history of VTE) who delivered | 1 July 2005 to 31 December 2011 | 30.32 | 3% | ▲ Novel (Sultan) ▲ RCOG§ ▲ SFOG (Swedish Guidelines) | Postpartum VTE (6 weeks) | 0.08% (validation cohort) |
| Tran et al, 2019[37] | USA | R, CS | Single | 6094 | All women who delivered after 14 weeks | 01 January 2015 to 31 December 2016 | NR | NR | ▲ RCOG ▲ Padua ▲ Caprini | Postpartum VTE (6 months) | 0.05% |
| Postpartum following caesarean delivery | | | | | | | | | | | |
| Binstock and Larkin, 2019 (abstract)[24] | USA | R, CS | Single | 2875 | Postpartum women following CD | 2011 | NR | NR | ▲ Novel (Binstock) ▲ RCOG | Postpartum VTE, not defined (NR) | 0.38% |
| Cavazza et al, 2012[25] | Italy | P, CS | Single | 501 | Postpartum women following CD | 2007 to 2009 | 34 | 53.5% | ▲ Novel (Cavazza) | Postpartum VTE, symptomatic, not defined (90 days) | 0.20% |
| Lok et al, 2019[33] | Hong Kong | P, CS | Single | 859 | Postpartum women following CD | May 2017 to April 2018 | 32.9 | 3.3% | ▲ Novel (Lok) ▲ RCOG ▲ ACOG | Postpartum VTE, symptomatic, not defined (NR) | Unable to estimate—no VTE |

*Retrospective case–control study of pregnant and postpartum women, but data reported for antepartum period only due to low number of postpartum VTE events (n=2).
†Internal validation study (ie, prediction model development without external validation).
‡Prospective cohort study with retrospective analysis, thus classified as retrospective cohort study.
§RCOG was applied to an English derivation cohort, n=433 353, incidence, 0.07% (312 events).
ACCP, American College of Chest Physicians; ACOG, American College of Obstetricians and Gynecologists; ASH, American Society of Hematology; BMI, body mass index; CC, case–control; CD, caesarean delivery; CS, cohort study; EThIG, Efficacy of Thromboprophylaxis as an Intervention during Gravidity Investigators; NR, not reported; NRS, non-randomised study; P, prospective; R, retrospective; RAM, risk assessment model; RCOG, Royal College of Obstetricians and Gynaecologists; SFOG, Swedish Society of Obstetrics and Gynecology; VTE, venous thromboembolism.

**Table 2** Summary of each study's risk of bias and applicability concern using the PROBAST (Prediction model Risk Of Bias ASsessment Tool)—review authors' judgements

| Author, year | Risk of bias | | | | Applicability | | | Overall | |
|---|---|---|---|---|---|---|---|---|---|
| | Participant selection | Predictors | Outcome | Analysis | Participant selection | Predictors | Outcome | Risk of bias | Applicability |
| Bauersachs et al, 2007[23] | ? | ? | + | – | ? | ? | + | – | ? |
| Binstock and Larkin, 2019[24] | ? | ? | ? | – | – | ? | ? | – | – |
| Cavazza et al, 2012[25] | – | ? | ? | – | – | + | ? | – | – |
| Chau et al, 2019[26] | ? | ? | ? | – | ? | ? | ? | – | ? |
| Chauleur et al, 2008[27] | ? | ? | ? | – | ? | ? | ? | – | ? |
| Dargaud et al, 2017[28] | ? | ? | ? | – | ? | ? | ? | – | ? |
| Dargaud et al, 2005[29] | – | ? | ? | – | ? | + | ? | – | ? |
| Ellis-Kahana et al, 2020[39] | – | ? | ? | – | ? | ? | ? | – | ? |
| Gassmann et al, 2021[30] | ? | ? | ? | – | ? | ? | ? | – | ? |
| Hase et al, 2018[31] | ? | ? | ? | – | – | ? | ? | – | – |
| Lindqvist et al, 2008[32] | – | ? | ? | – | ? | ? | ? | – | ? |
| Lok et al, 2019[33] | ? | ? | – | – | – | + | ? | – | – |
| Shacaluga and Rayment, 2019[34] | – | ? | ? | – | ? | ? | ? | – | ? |
| Sultan et al, 2016[35] | – | ? | + | + | + | ? | + | – | ? |
| Testa et al, 2015[36] | ? | ? | ? | – | ? | ? | ? | – | ? |
| Tran et al, 2019[37] | – | ? | ? | – | ? | ? | ? | – | ? |
| Weiss and Bernstein, 2000[38] | – | ? | ? | – | ? | ? | ? | – | ? |

+ indicates low risk of bias/low concern regarding applicability; –, indicates high risk of bias/high concern regarding applicability; and ? indicates unclear risk of bias/unclear concern regarding applicability.

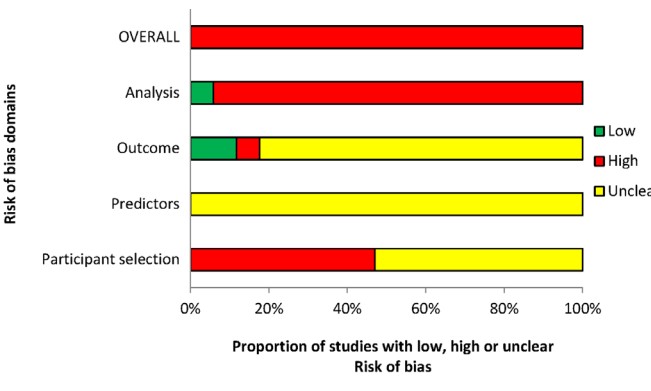

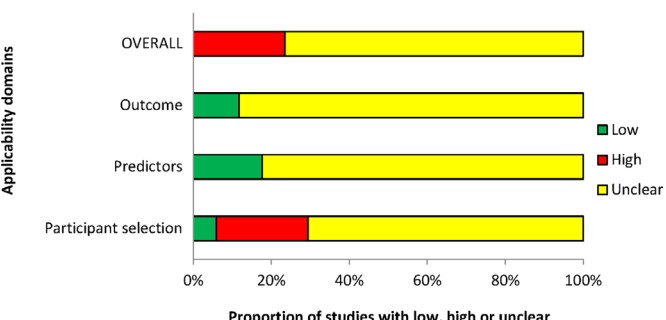

**Figure 2** PROBAST (Prediction model Risk Of Bias ASsessment Tool) assessment summary graph—review authors' judgements.

and without VTE in the external Swedish cohort with a C-statistic of 0.73 (95% CI: 0.71 to 0.75), and calibration, of observed and predicted VTE risk, close to ideal (calibration slope of 1.11 (95% CI: 1.01 to 1.20)). In the remaining studies, interpretation was further limited by marked heterogeneity, which was exacerbated when different thresholds were reported by different studies evaluating the same model. In general, model accuracy was generally poor, with high sensitivity usually reflecting a threshold effect, as indicated by corresponding low specificity values (and vice versa).

## DISCUSSION
### Summary of results
This systematic review identified 19 externally validated RAMs (and 1 internally validated risk model) that aimed to predict the risk of VTE in pregnant and postpartum women and who could be selected for thromboprophylaxis. Although various risk models (based on a variety of predictor variables) are being used, most of these lacked rigorous development and evaluation. The predictive accuracy of the RAMs was highly variable, and the substantial risk of bias concerns and the general lack of methodological clarity and unclear applicability make meaningful comparisons of the evidence difficult.

### Interpretation of results
Despite the development and use of various RAMs to predict the risk of developing VTE in women who are

pregnant or in the puerperium (within 6 weeks post-delivery), VTE remains the leading cause of direct maternity mortality in the UK (MBRRACE-UK report 2021). Several explanations for this are possible: the risk assessment tools are inadequate; the application of these tools is incomplete or inaccurate; the underlying VTE risks of the pregnant population (increasing age, body mass index and comorbidities) are changing from when the RAMS were developed; or all three problems are operating.

The use of thromboprophylaxis was reported in nine studies[23 25 28–31 33 35 36] (ranging from 3%[35] to 100%[23 28]). This may lead to underestimation of predictive accuracy if a given RAM was to predict VTE events that were subsequently prevented by thromboprophylaxis. In the remaining studies (n=8) where thromboprophylaxis use was not reported (n=8), further analysis of its impact on the performance of the RAMs was not possible. This also suggests that the degree to which thromboprophylaxis reduces the risk of VTE in those who received it cannot be accurately estimated. Moreover, the lack of data on the predictive performance of weight-based LMWH dosing, dosage change throughout pregnancy and D-dimer testing in the included studies also precluded further analysis of its association with VTE.

### Comparison to the existing literature
To our knowledge, there are no previous systematic reviews on this topic. However, recently several large registries have been interrogated in an attempt to derive robust prediction rules for this population, although with some methodological concerns. Sultan et al,[35] developed (using a large English-based registry database covering 6% of the population) and validated (using a Swedish national database registry) a risk prediction tool to estimate the absolute risk of VTE in postpartum women according to their individual risk factor combinations. Despite the low incidence of VTE in both cohorts (<0.08%), their model showed good discrimination in the external cohort and poor sensitivity at predicting those at risk of experiencing VTE. In addition, their model lacked some important VTE risk factors (eg, thrombophilia, antepartum immobilisation), and possibly underestimated the risks due to diagnosis limited to diagnostic coding (eg, varicose veins, severity of comorbidities) and the use of thromboprophylaxis in both cohorts.[42] Ellis-Kahana et al,[39] also derived (using a large national database from the USA) a risk prediction model for VTE in obese pregnant women and indicated strong discrimination. However, this model still requires external validation.

### Strengths and limitations
This systematic review has several strengths. It is the first systematic review to evaluate RAMs for predicting the risk of developing VTE in women during pregnant and the puerperium periods, and was conducted with robust methodology in accordance with the PRISMA statement[15] and the protocol was registered with the PROSPERO register. Clinical experts, in addition to the core review

**Table 3** Performance of RAMs applied antepartum to predict VTE

| Risk assessment models | Threshold or cut-off | Endpoint | Data source | Performance measures | | | | | |
| --- | --- | --- | --- | --- | --- | --- | --- | --- | --- |
| | | | | TP | FP | FN | TN | Sensitivity (95% CI) | Specificity (95% CI) |
| Predicting either antepartum or postpartum VTE | | | | | | | | | |
| All Wales (one study) | NR | VTE | Shacaluga and Rayment[34] | 25 | NR | 9 | NR | 0.74 (0.57 to 0.85) | NR |
| EThIG (one study) | High/very high risk | VTE | Bauersachs et al[23] | 5 | 580 | 0 | 225 | 1.00 (0.57 to 1.00) | 0.28 (0.25 to 0.31) |
| Lyon (two studies) | Risk score ≥3 | VTE | Dargaud et al[28] | 5 | 282 | 1 | 157 | 0.83 (0.44 to 0.97) | 0.36 (0.31 to 0.4) |
| Lyon | Risk score ≥3 | VTE | Dargaud et al[29] | 1 | 56 | 0 | 59 | 1.00 (0.21 to 1.00) | 0.51 (0.42 to 0.6) |
| RCOG (modified) (one study) | Risk score ≥3 | VTE | Hase et al[31] | 0 | 34 | 0 | 18 | unable to estimate – no VTE | 0.35 (0.23 to 0.48) |
| STRATHEGE (one study) | Risk score ≥3 | VTE | Chauleur et al[27] | 0 | 54 | 9 | 2622 | 0.00 (0.00 to 0.3) | 0.98 (0.97 to 0.99) |
| Testa 2015 (one study) | Risk score ≥2.5 | VTE | Testa et al[36] | 0 | 85 | 0 | 1634 | unable to estimate – no VTE | 0.95 (0.94 to 0.96) |
| Predicting antepartum VTE | | | | | | | | | |
| EThIG (one study) | High/very high risk | VTE | Bauersachs et al[23] | 2 | 583 | 0 | 225 | 1.00 (0.34 to 1.00) | 0.28 (0.25 to 0.31) |
| Lyon (one study) | Risk score ≥3 | VTE | Dargaud et al[28] | 1 | 286 | 1 | 157 | 0.50 (0.09 to 0.91) | 0.35 (0.31 to 0.4) |
| STRATHEGE (one study) | Risk score ≥1 | VTE | Chauleur et al[27] | 0 | 54 | 4 | 2627 | 0.00 (0.00 to 0.49) | 0.98 (0.97 to 0.99) |
| Weiss 2000 (one study) | Risk score ≥2 | VTE | Weiss and Bernstein[38] | 4 | 3 | 15 | 54 | 0.21 (0.09 to 0.43) | 0.95 (0.86 to 0.98) |
| Predicting postpartum VTE | | | | | | | | | |
| EThIG (one study) | High/very high risk | VTE | Bauersachs et al[23] | 3 | 582 | 0 | 225 | 1.00 (0.44 to 1.00) | 0.28 (0.25 to 0.31) |
| Lyon (one study) | Risk score ≥3 | VTE | Dargaud et al[28] | 4 | 283 | 0 | 158 | 1.00 (0.51 to 1.00) | 0.36 (0.31 to 0.4) |
| STRATHEGE (one study) | Risk score ≥1 | VTE | Chauleur et al[27] | 0 | 54 | 5 | 2626 | 0.00 (0.00 to 0.43) | 0.98 (0.97 to 0.98) |

EThIG, Efficacy of Thromboprophylaxis as an Intervention during Gravidity Investigators; FN, false negative; FP, false positive; NR, not reported; RAMs, risk assessment models; RCOG, Royal College of Obstetricians and Gynaecologists; TN, true negative; TP, true positive; VTE, venous thromboembolism.

**Table 4** Performance of RAMs applied postpartum to predict VTE

| Risk assessment models | Threshold or cut-off | Endpoint | Data source | Performance measures | | | | Sensitivity (95% CI) | Specificity (95% CI) |
|---|---|---|---|---|---|---|---|---|---|
| | | | | TP | FP | FN | TN | | |
| Predicting postpartum VTE following vaginal and caesarean delivery | | | | | | | | | |
| ACCP (one study) | NR | VTE | Gassmann et al[30] | 0 | 34 | 0 | 310 | unable to estimate – no VTE | 0.90 (0.86 to 0.93) |
| ACOG (one study) | NR | VTE | Gassmann et al[30] | 0 | 30 | 0 | 314 | unable to estimate – no VTE | 0.91 (0.88 to 0.94) |
| ASH (one study) | NR | VTE | Gassmann et al[30] | 0 | 0 | 0 | 344 | unable to estimate – no VTE | 1.00 (0.99 to 1.00) |
| Caprini (one study) | Risk score ≥2 | VTE | Tran et al[37] | 3 | 5780 | 0 | 311 | 1.00 (0.44 to 1.00) | 0.05 (0.05 to 0.06) |
| Caprini | Risk score ≥3 | VTE | Tran et al[37] | 1 | 3066 | 2 | 3025 | 0.33 (0.06 to 0.79) | 0.50 (0.48 to 0.51) |
| Caprini | Risk score ≥4 | VTE | Tran et al[37] | 0 | 1257 | 3 | 4834 | 0.00 (0.00 to 0.56) | 0.79 (0.78 to 0.80) |
| Padua (one study) | Risk score ≥4 | VTE | Tran et al[37] | 0 | 50 | 3 | 6041 | 0.00 (0.00 to 0.56) | 0.99 (0.99 to 0.99) |
| RCOG (three studies) | NR | VTE | Gassmann et al[30] | 0 | 138 | 0 | 206 | unable to estimate – no VTE | 0.60 (0.55 to 0.65) |
| RCOG | Risk score ≥2 | VTE | Tran et al[37] | 1 | 3837 | 2 | 2254 | 0.33 (0.06 to 0.79) | 0.37 (0.36 to 0.38) |
| RCOG | ≥2 low risk factors or 1 high risk factor | VTE | Sultan et al[35] | 197 | 149205 | 115 | 283836 | 0.63 (0.58 to 0.68) | 0.66 (0.65 to 0.66) |
| SFOG (two studies) | Risk score ≥2 | VTE | Lindqvist et al[32] | 18 | 111 | 19 | 2273 | 0.49 (0.33 to 0.64) | 0.95 (0.94 to 0.96) |
| SFOG | ≥2 risk factors | VTE | Sultan et al[35] | 109 | 41145 | 412 | 620721 | 0.21 (0.18 to 0.25) | 0.94 (0.94 to 0.94) |
| Chau, 2019 (one study*) | Risk score ≥3 (2012 data set) | VTE | Chau et al[26] | 0 | 101 | 1 | 456 | 0.00 (0.00 to 0.79) | 0.82 (0.78 to 0.85) |
| Chau, 2019 | Risk score ≥3 (2015 data set) | VTE | Chau et al[26] | 0 | 113 | 1 | 393 | 0.00 (0.00 to 0.79) | 0.78 (0.74 to 0.81) |
| Ellis-Kahana, 2020 (full model) (one study†) | Risk score >3 (high risk) | VTE | Ellis-Kahana et al[39] | 68 | 7942 | 41 | 75449 | 0.62 (0.53 to 0.71) | 0.90 (0.90 to 0.91) |
| Ellis-Kahana, 2020 (without antepartum thromboembolic disorder) | Risk score >3 (high risk) | VTE | Ellis-Kahana et al[39] | 63 | 9926 | 46 | 73465 | 0.58 (0.48 to 0.67) | 0.88 (0.88 to 0.88) |
| Sultan, 2016 (one study‡) | ≥2 risk factors: top 35% (threshold: 7.2 per 10 000 deliveries) | VTE | Sultan et al[35] | 355 | 231480 | 166 | 430386 | 0.68 (0.64 to 0.72) | 0.65 (0.65 to 0.65) |
| | ≥2 risk factors: top 25% (threshold: 8.7 per 10 000 deliveries) | VTE | Sultan et al[35] | 310 | 164976 | 211 | 496890 | 0.60 (0.55 to 0.64) | 0.75 (0.75 to 0.75) |
| | ≥2 risk factors: top 20% (threshold: 9.8 per 10 000 deliveries) | VTE | Sultan et al[35] | 278 | 131921 | 243 | 529945 | 0.53 (0.49 to 0.58) | 0.80 (0.80 to 0.80) |
| | ≥2 risk factors: top 10% (threshold: 14 per 10 000 deliveries) | VTE | Sultan et al[35] | 185 | 66053 | 336 | 595813 | 0.36 (0.32 to 0.40) | 0.90 (0.90 to 0.90) |
| | ≥2 risk factors: top 6% (threshold: 18 per 10000 deliveries) | VTE | Sultan et al[35] | 158 | 41096 | 363 | 620770 | 0.30 (0.27 to 0.34) | 0.94 (0.94 to 0.94) |
| | ≥2 risk factors: top 5% (threshold: 19.7 per 10 000 deliveries) | VTE | Sultan et al[35] | 139 | 32980 | 382 | 628886 | 0.27 (0.23 to 0.31) | 0.95 (0.95 to 0.95) |
| | ≥2 risk factors: top 1% (threshold: 41.2 per 10 000 deliveries) | VTE | Sultan et al[35] | 47 | 6576 | 474 | 655290 | 0.09 (0.07 to 0.12) | 0.99 (0.99 to 0.99) |
| Predicting postpartum VTE following caesarean delivery only | | | | | | | | | |
| ACOG (one study) | Risk score ≥3 | VTE | Lok et al[33] | 0 | 0 | 0 | 859 | unable to estimate – no VTE | 1.00 (1.00 to 1.00) |
| RCOG (two studies) | NR | VTE | Binstock and Larkin (abstract)[24] | 11 | 2692 | 0 | 172 | 1.00 (0.74 to 1.00) | 0.06 (0.05 to 0.07) |
| RCOG | Risk score ≥3 | VTE | Lok et al[33] | 0 | 649 | 0 | 210 | unable to estimate – no VTE | 0.24 (0.22 to 0.27) |

Continued

**Table 4** Continued

| Risk assessment models | Threshold or cut-off | Endpoint | Data source | Performance measures | | | | Sensitivity (95% CI) | Specificity (95% CI) |
|---|---|---|---|---|---|---|---|---|---|
| | | | | TP | FP | FN | TN | | |
| Binstock, 2019 (one study) | NR | VTE | Binstock and Larkin (abstract)[24] | 11 | 2635 | 0 | 229 | 1.00 (0.74 to 1.00) | 0.08 (0.07 to 0.09) |
| Cavazza, 2012 (one study) | Moderate/high/very high | VTE | Cavazza et al[25] | 0 | 268 | 1 | 232 | 0.00 (0.00 to 0.79) | 0.46 (0.42 to 0.51) |
| Lok, 2019 (one study) | Risk score ≥3 | VTE | Lok et al[33] | 0 | 28 | 0 | 831 | unable to estimate – no VTE | 0.97 (0.95 to 0.98) |

*Data discrepancy in paper—text states analysis included 1069 women: 557 in the 2012 time frame and 512 in the 2015 time frame; however, data in tables suggest 558 women included in the 2012 time frame and 507 in the 2015 time frame.
†Internal validation study. Full risk prediction model: C-statistic, 0.817 (95% CI: 0.768 to 0.865) with Hosmer-Lemeshow p value=0.297; model without antepartum thromboembolic disorder: C-statistic, 0.778 (95% CI: 0.729 to 0.826) with Hosmer-Lemeshow p value=0.114.
‡Sultan et al,[35] final risk prediction model in external Swedish cohort: C-statistic, 0.73 (95% CI: 0.71 to 0.75) and calibration slope, 1.11 (95% CI: 1.01 to 1.20).
ACCP, American College of Chest Physicians; ACOG, American College of Obstetricians and Gynecologists; ASH, American Society of Hematology; FN, false negative; FP, false positive; NR, not reported; RAMs, risk assessment models; RCOG, Royal College of Obstetricians and Gynaecologists; SFOG, Swedish Society of Obstetrics and Gynecology; TN, true negative; TP, true positive; VTE, venous thromboembolism.

team, were involved and consulted throughout as advisors and to assess the validity and applicability of research findings during the review processes.

The main limitations of this study related to the observational nature of the studies reviewed and their own limitations. Most of the included risk prediction studies were retrospective cohorts. Retrospective cohort studies of large health database registries are limited by poor data quality and failure to accurately ascertain outcomes and case–control designs are prone to bias including uncontrolled confounding, temporal and selection bias.[43] Conversely, better quality data may be obtained with prospective cohorts, but smaller sample sizes will lack statistical power. In addition, most of the external validation studies evaluated predictive performance of risk models that were not statistically derived (ie, without model development and internal validation). This process is vital, as risk models with only external validation may be subject to overfitting and optimism.[40] Similarly, the absence of model performance measures such as calibration or discrimination hinders the full appraisal of models.[41]

Due to the high levels of heterogeneity between studies, we were unable to undertake any meta-analysis or statistical examination of the causes of heterogeneity due to the small number of external validation studies per risk model. Potential sources of heterogeneity include variation in study design, the study population, risk model implementation, outcome definition and measurement and the use of thromboprophylaxis. As a result, we reported descriptive statistics to provide a better understanding of the evidence base applicable to the subject matter, and shortcomings regarding reliability and validity of the data. Finally, assessments on study relevance, information gathering and validity of articles were unblinded and could potentially have been influenced by preformed opinions. However, masking is resource intensive with uncertain benefits in protecting against bias decisions.[44]

### Implications for policy, practice and future research

VTE risk assessment is challenging for numerous reasons. Many risk factors for VTE are pre-existing and non-modifiable (such as parity and inherited thrombophilia). These are then often combined with evolving risk factors which can change over the course of a pregnancy or postnatal period. Despite wide scale awareness of VTE being a major contributor to maternal mortality, numerous challenges with VTE risk stratification have been highlighted. In the UK, the MBRRACE-UK report (Saving Lives, Improving Mothers' Care 2018)[45] shows that doctors and midwives find existing risk scoring systems difficult to apply consistently in clinical practice. There is a need for development of an RAM that is simpler and more reproducible. National Institute for Health and Care Excellence guidelines on the use of thromboprophylaxis (NG89)[46] concluded that the tool described by Sultan et al[35] showed poor sensitivity compared with their prespecified target of 90% sensitivity. However, this high level of sensitivity

may not be realistic because there is evidence that only 70% of women having antenatal pulmonary embolism had any identifiable classic risk factors suggesting that sensitivity rates above 70% may not be achievable.[47] In addition, a high sensitivity rate is usually associated with a lower specificity rate and the overall balance of benefits and harms may be undesirable if that means exposing a high proportion of women to thromboprophylaxis.

Despite lack of evidence, many guidelines and clinical care bundles include the use of RAMs to guide VTE prophylaxis. Recently published ACOG guidelines state that most RAMs have not been validated prospectively in the obstetrical population and that current usage of such models is based on extrapolations from non-pregnant women, who differ biologically from pregnant women. The practice bulletin emphasises the need for more research to identify optimal models.[37] Although further research is clearly needed the routine use of thromboprophylaxis may present a barrier to generating accurate and precise estimates of the prognostic accuracy of RAMs. Further work to improve RAMs to help stratify the risk of VTE in women who are pregnant or in the puerperium could focus on using decision-analytical modelling to compare the effects, harms and costs of giving thromboprophylaxis to patients with varying risks of VTE. This would allow determination of the risk threshold at which thromboprophylaxis provides optimal overall benefit. Subsequent work to validate these findings would require primary research. Despite the limitations of undertaking accuracy studies in populations where thromboprophylaxis is routinely used, future research could focus on selected higher risk groups who are more likely to benefit from prophylaxis and, with a higher prevalence of VTE, are more amenable to an appropriately powered prospective study. However, given the uncertain benefits and harms of VTE thromboprophylaxis during pregnancy and the postpartum period,[14 48] risk prediction studies should be undertaken alongside (or as a part of) randomised trials of prophylaxis in targeted groups deemed to be at higher risk of VTE.

## CONCLUSIONS

Currently, there are a number of risk assessment models for assessing risk of VTE in pregnancy and the puerperium. Our review has shown that none of these models has been adequately validated and they have limited abilities to detect those at risk of VTE.

**Acknowledgements** The authors would like to thank all additional members of the core project group for NIHR131021 for input and commentary throughout the work. We are also indebted to Donna Davis for assistance with logistics and administration.

**Contributors** AP coordinated the review. SD, AP, JD, JH, MC, BJH, CN-P and SG were responsible for the conception, design and obtaining funding for the study. MC developed the search strategy, undertook searches and organised the retrieval of papers. AP, GR, JH, BJH, JD, CN-P and SG were responsible for the acquisition, analysis and interpretation of data. BJH, JD, CN-P and SG helped interpret and provided a methodological, policy and clinical perspective on the data. AP, BJH, JD and SD were responsible for the drafting of this paper, although all authors provided comments on the drafts, read and approved the final version. AP is the guarantor for the paper.

**Funding** This study was funded by the UK National Institute for Health Research Health Technology Assessment (NIHR HTA) Programme (project number 131021). The views expressed in this paper are those of the authors and not necessarily those of the NIHR HTA Programme. Any errors are the responsibility of the authors. The funders had no role in the study design, in the collection, analysis and interpretation of data; in the writing of the manuscript; and in the decision to submit the manuscript for publication.

**Competing interests** CN-P reports personal fees from Sanofi, and is the lead developer of the RCOG Green Top Guideline on thromboprophylaxis in pregnancy (37a). All other authors declare no competing interests.

**Patient and public involvement** Patients and/or the public were not involved in the design, or conduct, or reporting, or dissemination plans of this research.

**Patient consent for publication** Not applicable.

**Ethics approval** Not applicable.

**Provenance and peer review** Not commissioned; externally peer reviewed.

**Data availability statement** All data relevant to the study are included in the article or uploaded as supplementary information.

**ORCID iDs**
Abdullah Pandor http://orcid.org/0000-0003-2552-5260
Jahnavi Daru http://orcid.org/0000-0001-5816-2609
Beverley J Hunt http://orcid.org/0000-0002-4709-0774
Gill Rooney http://orcid.org/0000-0002-8388-9444
Jean Hamilton http://orcid.org/0000-0003-3326-9842
Mark Clowes http://orcid.org/0000-0002-5582-9946
Steve Goodacre http://orcid.org/0000-0003-0803-8444
Catherine Nelson-Piercy http://orcid.org/0000-0001-9311-1196
Sarah Davis http://orcid.org/0000-0002-6609-4287

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
