## [Reviewer comments · BMJ Open]

ARTICLE DETAILS

TITLE (PROVISIONAL)	Risk assessment models for venous thromboembolism in pregnancy and in the puerperium: a systematic review
AUTHORS	Pandor, Abdullah; Daru, Jahnavi; Hunt, Beverley; Rooney, Gill; Hamilton, Jean; Clowes, Mark; Goodacre, Steve; Nelson-Piercy, Catherine; Davis, Sarah

VERSION 1 – REVIEW

REVIEWER	Lucia Stanciakova University Hospital Martin, National Centre of Haemostasis and Thrombosis, Department of Haematology and Blood Transfusion
REVIEW RETURNED	05-Jul-2022

GENERAL COMMENTS	I would like to sincerely thank the authors for the comprehensive review with taking into account a relatively high number of evaluated parameters. Please, do not consider the following questions badly - I only suppose that the answers to them may be interesting for the assessment of the clinical management and further evaluation of the data from the manuscript: 1. Can the authors add the data about the dosage of LMWH and correlate it with the mean weight at the beginning of pregnancy and before the delivery ?2. Can they specify in which gestational week the need of the change in dosage of LMWH was present ?3. Do the authors have any information about D-dimer levels in the included patients and their dynamics in the association with developed VTE ? D-dimers are considered to be non-specific parameters indicating the risk of VTE...4. Do the authors have the information about adverse events of the LMWH used in the included patients – e.g. incidence of bleeding events, allergic reaction...?5. From the practical point of view, can the authors make any suggestion for the recommendations for the clinical practice based on the results of this study ? From my point of view, the article can be published after incorporation of the answers to the questions and after such a minor revision.
--

REVIEWER	Victoria Bitsadze I M Sechenov First Moscow State Medical University
REVIEW RETURNED	13-Jul-2022

GENERAL COMMENTS	The review highlights extremely important issues of modern obstetrics -- venous thromboembolism and antithrombotic prophylaxis during pregnancy and the postpartum period . Despite
---

	the use of various RAMs in clinical practice, VTE prevention is still far from the desired level. This comparative efficacy study of the currently proposed PAMs is the first meta-analysis performed at a high methodological level. The design of the study, the competent use of statistical processing of the data obtained and the discussion allowed the authors to draw reasonable conclusions. The results of the study indicate the need for further work to improve and create an effective and at the same time easily applicable in clinical practice RAM. Congratulations to the authors on a great job!
--	--

VERSION 1 – AUTHOR RESPONSE

Reviewer #1 I would like to sincerely thank the authors for the comprehensive review with taking into account a relatively high number of evaluated parameters. Please, do not consider the following questions badly - I only suppose that the answers to them may be interesting for the assessment of the clinical management and further evaluation of the data from the manuscript:	Thank you for your in-depth review, and your positive comments on the conduct, standard and writing of our work.
1. Can the authors add the data about the dosage of LMWH and correlate it with the mean weight at the beginning of pregnancy and before the delivery?	Thank you for this suggestion. Although our review specifically focused on evaluating the comparative accuracy of risk assessment models for predicting the risk of developing VTE in women who are pregnant or in the puerperium (within 6 weeks post-delivery) we were unable to undertake these additional post-hoc analyses as relevant data were not reported in any of the included studies (e.g. mean LMWH dose, and mean weight at the beginning of pregnancy and before delivery). A sentence has been added in the discussion section to reflect this limitation.
2. Can they specify in which gestational week the need of the change in dosage of LMWH was present ?	Thank you for this suggestion. Unfortunately, we are unable to add this information to the review as this information is not reported in any of the included studies. A sentence has been added in the discussion section to reflect this limitation.

3. Do the authors have any information about D-dimer levels in the included patients and their dynamics in the association with developed VTE ? D-dimers are considered to be non-specific parameters indicating the risk of VTE...	Thank you for raising this important point. Unfortunately, none of the included studies provide information on D-dimer testing and its association with VTE. A sentence has been added in the discussion section to reflect this limitation.
4. Do the authors have the information about adverse events of the LMWH used in the included patients – e.g. incidence of bleeding events, allergic reaction...?	Thank you for raising this important point. As our review specifically focused on evaluating the comparative accuracy of risk assessment models for predicting the risk of developing VTE in women who are pregnant or in the puerperium (within 6 weeks post-delivery), information on adverse events due to the use of LMWH is poorly reported in 9 out of 17 studies (e.g. adverse event outcomes are poorly defined, heterogeneous and inconsistently reported). Whilst we would be happy to add this information to our review appendix (see table below) if required by the peer reviewer, we believe this has been more comprehensively reviewed by Sirico et al. and Lu et al. Sirico A, Saccone G, Maruotti GM, Grandone E, Sarno L, Berghella V, Zullo F, Martinelli P. Low molecular weight heparin use during pregnancy and risk of postpartum hemorrhage: a systematic review and meta-analysis. J Matern Fetal Neonatal Med. 2019 Jun;32(11):1893-1900. doi: 10.1080/14767058.2017.1419179. Lu E, Shatzel JJ, Salati J, DeLoughery TG. The Safety of Low-Molecular-Weight Heparin During and After Pregnancy. Obstet Gynecol Surv. 2017 Dec;72(12):721-729. doi: 10.1097/OGX.0000000000000505. PMID: 29280473.
5. From the practical point of view, can the authors make any suggestion for the recommendations for the clinical practice based	Thank you for this suggestion. We believe we have made appropriate research recommendations based on the findings from this

n the results of this study ?	study. Given the uncertain benefits and harms of VTE thromboprophylaxis during pregnancy we have recommended further work to improve RAMs and undertake risk prediction studies alongside (or as a part of) randomised trials of prophylaxis in targeted groups deemed to be at higher risk of VTE.
Reviewer #2 The review highlights extremely important issues of modern obstetrics -- venous thromboembolism and antithrombotic prophylaxis during pregnancy and the postpartum period. Despite the use of various RAMs in clinical practice, VTE prevention is still far from the desired level. This comparative efficacy study of the currently proposed PAMs is the first meta-analysis performed at a high methodological level. The design of the study, the competent use of statistical processing of the data obtained and the discussion allowed the authors to draw reasonable conclusions. The results of the study indicate the need for further work to improve and create an effective and at the same time easily applicable in clinical practice RAM. Congratulations to the authors on a great job!	Thank you for your in-depth review, and your positive comments on the conduct, standard and writing of our work. We agree this topic is important, very relevant and timely.

VERSION 2 – REVIEW

REVIEWER	Lucia Stanciakova University Hospital Martin, National Centre of Haemostasis and Thrombosis, Department of Haematology and Blood Transfusion
REVIEW RETURNED	14-Sep-2022
GENERAL COMMENTS	I would like to kindly thank the authors for the response to the comments of the reviewers. I suppose that the obscure points became clearer. Therefore, from my point of view, the manuscript can be published in the actual form.